# Heterocyclic Crown Ethers with Potential Biological and Pharmacological Properties: From Synthesis to Applications

Faiz Ullah [1], Taskin Aman Khan [2], Jawaria Iltaf [2], Saleha Anwar [3], Muhammad Farhan Ali Khan [4], Muhammad Rizwan Khan [5], Sami Ullah [6], Muhammad Fayyaz ur Rehman [2], Muhammad Mustaqeem [7], Katarzyna Kotwica-Mojzych [8] and Mariusz Mojzych [9],*

1   Department of Chemistry, Quaid I Azam University, Islamabad 45320, Pakistan; faizullah@chem.qau.edu.pk
2   Institute of Chemistry, University of Sargodha, Sargodha 40100, Pakistan; taskinaman5@gmail.com (T.A.K.); javeriaaltaf71@gmail.com (J.I.); fayyaz9@gmail.com (M.F.u.R.)
3   Department of Chemistry, International Islamic University, Islamabad 45320, Pakistan; Salehanwar1895@gmail.com
4   Department of Pharmacy, Quaid I Azam University, Islamabad 45320, Pakistan; farhanali@bs.qau.edu.pk
5   Department of Chemistry, Allama Iqbal Open University, Islamabad 45320, Pakistan; Rizwankhan_2200@yahoo.com
6   Department of Zoology, Government College University, Faisalabad 38000, Pakistan; samibhatti925@gmail.com
7   Department of Chemistry, Sub Campus, University of Sargodha, Bhakkar 30000, Pakistan; Muhammad.mustaqeem@uos.edu.pk
8   Department of Histology, Embryology and Cytophysiology, Medical University of Lublin, Radziwiłłowska 11, 20-080 Lublin, Poland; katarzynakotwicamojzych@umlub.pl
9   Department of Chemistry, Siedlce University of Natural Sciences and Humanities, 3-go Maja 54, 08-110 Siedlce, Poland
*   Correspondence: mariusz.mojzych@uph.edu.pl

**Abstract:** Cyclic organic compounds with several ether linkages in their structure are of much concern in our daily life applications. Crown ethers (CEs) are generally heterocyclic and extremely versatile compounds exhibiting higher binding affinity. In recent years, due to their unique structure, crown ethers are widely used in drug delivery, solvent extraction, cosmetics manufacturing, material studies, catalysis, separation, and organic synthesis. Beyond their conventional place in chemistry, this review article summarizes the synthesis, biological, and potential pharmacological activities of CEs. We have emphasized the prospects of CEs as anticancer, anti-inflammatory, antibacterial, and antifungal agents and have explored their amyloid genesis inhibitory activity, electrochemical, and potential metric sensing properties. The central feature of these compounds is their ability to form selective and stable complexes with various organic and inorganic cations. Therefore, CEs can be used in gas chromatography as the stationary phase and are also valuable for cation chromatographic to determine and separate alkali and alkaline-earth cations.

**Keywords:** crown ethers; iontophoresis; heterocyclic compounds; macrocyclic polyether

## 1. Introduction

Crown ethers (CEs) are macrocyclic polyethers and have three to twenty oxygen atoms alienated by two or more carbon atoms. These macromolecules can be either substituted or unsubstituted. This class of organic compounds has an interesting structure with a hydrophobic ring surrounding a hydrophilic cavity [1]. Crown ethers are heterocyclic compounds present as cyclic oligomers in their simple form. These are extremely versatile compounds exhibiting higher binding affinity towards metal ions, including s-block and transition metal ions [2]. For example, 18-crown-6 has a cavity that fits the size of 4f transition metal ions and has reflected exceptional attraction for complexation with the lanthanide ions [3]. Ethyleneoxy moiety ($CH_2CH_2O$-) is a crucial repeated unit of simple crown ether: repeated twice in dioxane and six times in 18-crown-6 [4].

These can be either substituted or unsubstituted and possess a hydrophobic ring surrounding a hydrophilic cavity, enabling them to form stable complexes with metal ions and contributing to host-guest chemistry [1,5]. Since their discovery, they have an outstanding capability to selectively coordinate ions, making them attractive for broad research applications [6]. The binding ability of crown ethers either with organic molecules or ions depends on their cavity size. They can carry different ions in a non-aqueous solvent. Crown ether also acting as a phase transfer catalyst. For example, 18-crown-6 shows binding ability with $K^+$ ion, and 12-crown-4 tends to complex with $Li^+$. Owing to this unique ability, CEs have broad applications in chemistry, biotechnology, and biochemistry. Alkali metal complexes with macrocyclic ligands mediate electron transfer processes.

$$2M(S) + L \rightarrow M^+ L + M^- \tag{1}$$

where M is the alkali metal, *e.g.*, potassium or sodium, and L is the complexant, *e.g.*, 18-crown-6 as shown in Scheme 1 [7].

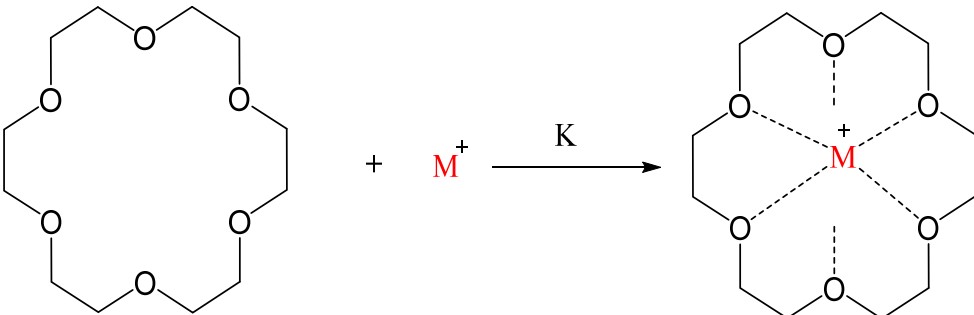

**Scheme 1.** Binding of $M^+$ by 18-crown-6 ether; K is complex stability constant [8].

The complex compounds, in which metal cation acts as guest and crown ether acts as the host, fall into the class of host-guest chemistry. Complex properties and manufacturing of crown ether were first studied by Pedersen in 1967 [9,10]. According to Pedersen's proposal, it is the electrostatic force of attraction between the metal cation and the oxygen atoms of the crown ether ring (host-guest complexes). Crown ethers behave as ligands in host-guest complexes and have a much weaker attraction for transition-metal cations than for alkali and alkaline earth metal cations [5]. This class acts as chiral selectors for the enantiomeric separation of different chiral compounds. Polyether (18-crown-6) has a strong binding ability with protonated chiral primary amines [11]. CEs change their color when they bind with an ion like sodium. There's an option that it can be engaged to make ion detectors. Nanoparticle improved with crown ethers can be used as metal ion sensors charged molecules in colorimetric methods [12]. In the name of CEs, the first number refers to the number of atoms in the ring, and the second number represents atoms of oxygen. The oxygen atom behaves as a Lewis base. Hexaether is the first CE that was accidentally found during the synthesis of bisphenol [5]. Crown ether and its derivatives have also been reported to show various biological properties [1]. Crown ether and cryptand complexes of alkali metals show tremendously low ionization potentials (1.52 eV) and act as super-alkalis. This innovation opens a novel track in designing chemical species with record low ionization potentials and synthesizing multiple charged Zintl clusters [13]. The first compound was discovered as dibenzo-18-crown-6 (0.4%) during the synthesis of the phenolic ligand from catechol and bis(2-chloroethyl)ether. It was 18-carbon atoms cyclic ring with 6-oxygen atoms and complexed with sodium cation [14].

Various simple crowns showed an ability to interact with enzymes, such as R-chymotrypsin, lipases, and subtilisin Carlsberg, benefited by the presence of these CEs [15]. The CEs came into the limelight due to their unique structure and applications in organic synthesis and drug delivery [16]. Ionophoric properties allow CEs to transport through membranes and interact with the living system [17]. Stability constants can be im-

proved by selecting CEs for various ions and via an extraction method [18]. Polyfunctional crown compounds act as photo-controlled ionophores or optical molecular sensors. The crowns have been entirely used to produce ion sensors that identify the presence of target ions [8,19]. Crown ethers are specifically designed and integrated for their interaction with DNA [20]. By using crown ether compounds, DNA intercalation and binding mechanisms have been studied [21,22].

In addition to the biological and catalytical activities, crown ethers also show toxicity. For example, 18-crown-6 can absorb through the skin and cause harmful effects on Central Nervous System (CNS) [23]. In terms of toxicity, simple CEs of 18-crown-6 were found less toxic than aspirin [24,25]. The CEs derivatives with the least toxicity can be synthesized to inhibit cell proliferation and kill pathogens [26]. The CEs analogs are also widely used in chromatography procedures [27] and during capillary electrophoresis. The CEs resins are used especially for chiral compounds [28]. They proved their significance in ion-exchange chromatography during both mobile and stationary phases [29]. Modern studies highlight different uses of CEs derivatives; fluorescent probes [30–32] chemosensing of bioactive species [33], chemo-sensors for bioactive molecular detection [34,35].

## 2. The History of Crown Ethers

Macrocyclic polyethers are crown ethers; they possess various ethylene oxide units covalently linked in a macrocyclic ring either substituted or unsubstituted and form the simplest form of cyclic oligomers of dioxane [36]. In 1967, Charles Pedersen prepared a crown ether to synthesize a complexing agent that could bind-divalent cations [37]. Two catecholate groups were linked through one hydroxyl. It reveals that the polydentate ligand partially envelopes the cation through phenolic hydroxyls ionization and neutralizes the bound dictation. Pederson found that this isolated by-product forms a potent complex with K$^+$ ions. The cyclic polyethers are complexing agents with a unique tendency to bind alkali metal cations [38,39]. Pedersen got fame through his dibenzo crown ethers work. Crown ether strong footed the areas of phase transfer catalysts, organic synthesis, and other disciplines [40]. In 1987, Charles Pedersen, Jean-Marie, and Donald Cram Lehn were awarded the Nobel Prize for chemistry because of their research on crown ethers and cryptands. Their work led to developing a new area of chemistry, such as supramolecular chemistry or macrocyclic [37]. Macrocyclic chemistry deals with macrocycle interactions with various metal ions. After Pedersen's crown ethers discovery, numerous macrocycles were synthesized, and their ability to form complexes was investigated [10]. Over the past years, great efforts have been made to synthesize crown ethers because of their application, especially regarding the drug delivery process [41].

The central feature of these compounds is the ability to form selective and stable complexes with various organic and inorganic cations. Before Pedersen's synthesis, a few examples of macrocycles were known [42]. He used nucleophiles such as resorcinol (1,3-dihydroxybenzene) and reacted with various substituted diol derivatives, resulting in the formation of various macrocyclic polyethers in the presence of Lewis's acid catalyst. The condensation of acetone and furan led to the formation of furan-acetone tetramer as shown in Scheme 2.

Initially, these compounds were named anhydrous tetramers because of the loss of four water molecules in the reaction [43].

In 1957, Borrows, Wadden, and Stewart treated an oxirane with alkyl aluminum, magnesium, and zinc to produce dioxane and various cyclic compounds; cyclic tetramer of ethylene oxide was one of them as shown in Figure 1 [44]. In the year 1959, Wilkinson et al., synthesized a cyclic tetramer using propylene oxide. At that time potential of the cyclic compounds was not appreciated [45].

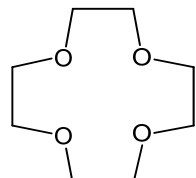

Furan + Acetone

Furan-acetone tetramer

**Scheme 2.** Formation of furan-acetone tetramer.

**Figure 1.** Cyclic tetramer of ethylene oxide.

    2,2′-Dichlorodiethyl ether reaction with mono-protected catechol using sodium hydroxide in *n*-butanol and after deprotection results in the phenolic derivative. Catechol that was unprotected in the initial mixture gave rise to the crown ether. Pedersen found that in the presence of sodium hydroxide, the solubility of dibenzo-18-crown-6 as shown in Figure 2. increased in methanol. It was due to the complexation between the crown ether and the sodium ion. Pedersen also showed that oxygen could be replaced by nitrogen and sulfur; likewise, other species can coordinate to the electron-rich compounds. Cram et al. introduced host-guest complexation, which describes the nitration between the host (crown ether) and the guest (metal ion) [10]. Compounds like crown ethers now became a central structure in supramolecular (host/guest) chemistry [46].

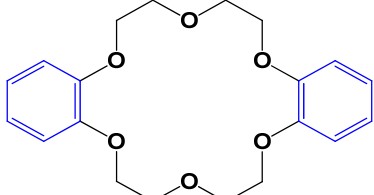

**Figure 2.** Structure of dibenzo-18-crown-6.

    Since dibenzo-18-crown-6 synthesis, various macrocycles are prepared, and their effect on host-guest interactions and separation science is widely studied. It helped to understand their properties as well as ionophores and crown ethers, which are biologically significant. Nonactin, a macrotetrolide antibiotic, is a naturally present ionophore whose principles of binding and transport cation abilities are now better understood through crown ether [47]. In addition to nonactin, there is another important biologically active compound, Valinomycin. This ionophore is highly flexible and possesses multiple donors.

The introduction of biological systems into the crown ether structure has provided scientists with an even greater insight into biological ionophores, such as cyclic peptides and macrolide antibiotics [48]. Alternatively, there is a targeted synthesis of macrocycles to make them resemble biological systems. Through this capable mimicking of macromolecular and various aspects, biological systems are possible [49]. The introduction of chirality in a macrocycle possesses multiple interaction sites, which make them attractive chiral receptors. That is why carbohydrate structure is incorporated in the crown ether framework [50].

Recently, the spirochetal ring system has been investigated. The structure and cation binding properties of monensin (acyclic ionophore) as shown in Figure 3. have significant importance regarding spiroacetal ionophores. They are used in the synthesis of cyclized lactone derivatives [10].

**Figure 3.** Structure of monensin.

*Crown Ethers as Solvent Extraction Reagents*

After the discovery, besides many uses of crown ethers, there was an important use as solvent extraction in which organopillic salt used was picrate. In 1962, dibenzo-18-crown-6 was found that formed an alkali-metal-salt adduct. This adduct was found to be soluble in organic solvents. Crystals of KMnO4, for example, may be induced to dissolve in aromatic hydrocarbons simply by adding perhydrodibenzo [18]-crown-6. This was the time when solvent extraction properties of crown ethers were discovered. Later, it was use for the complex formation properties of crown ether with metal ions. Picrate salts were used widely for this purpose. Since polyether complexes are soluble in certain organic solvents, salts may frequently be recovered from aqueous solutions into polyether-containing organic solvents. Such research has helped to demonstrate the crown ethers' selectivity and general behavior, notably in terms of complexation with alkali and alkaline earth metals. Similarly, various parallel studies have been reported in which picrate and chloroform was used as an anion and carrier diluent respectively. Solvent extraction and ion exchange scientists have been drawn to these investigations as well [51,52].

## 3. Synthesis of Crown Ethers and Their Derivatives

### 3.1. Synthesis of Dibenzo-18-crown-6 Ether

An innovative compound was synthesized as crown ether-functionalized benzimidazole in a formylation process started by dibenzo-18-crown-6 formylation with hexamethylenetetramine where trifluoroacetic acid was used as a catalyst. The use of trifluoroacetic acid provided a high yield (90%) of the final product, 4,4′-di-formyl dibenzo-18-crown-6. Due to this reason, methane sulphonic acid was replaced by trifluoroacetic acid as a former catalyst causing a lower yield, subsequently, no product isolation. The condensation of *o*-phenylenediamines with diformyl derivative or middle product leads to the formation of resultant benzimidazoles accompanied by the catalyst *p*-toluene sulphonic acid at lower reaction temperature and smaller reaction time (Scheme 3) [53].

**Scheme 3.** Preparation of dibenzo-18-crown-6 ether functionalized benzimidazoles.

## 3.2. Synthesis of Bis-benzo-15-crown-5 Ether

This synthesis was initiated when mixture of compound 1,2-bis(bromomethyl)benzene and ethanol solution reacted with a mixture of hydroxyl benzldehydes and sodium hydroxide solution. This reaction synthesized the main reactant formyl-substituted compounds for the final formation of biscrown ethers. These formyl substituted compounds were mixed and continuously stirred with 4′-aminobenzo-15-crown-5 ether and methanol. This reaction continued for 2 h at room temperature. The final product as bis-benzo-15-crown-5 ether was obtained and recrystallized from methanol as shown in Scheme 4 [54].

**Scheme 4.** Formation of bis-benzo-15-crown-5 ether as contained in Schiff bases.

### 3.3. Synthesis of Aza-Crown Ether-Squaramide Conjugates

This crowded Scheme 5 presents the formation of compounds (**8**–**11**) [55]. As reported earlier, an important structure 1,10-diaza-18-crown-6 was used as a starting material for the synthesis of compounds 2,2′-(1,4,10,13-tetraoxa-7,16-diazacyclooctadecane-7,16-diyl)diethanamine (**2**) and 2-(16-ethyl-1,4,10,13-tetraoxa-7,16-diazacyclo-octadecan-7-yl)ethaneamine (**5**) [56]. The product formation involved the reaction between 1,10-diaza-18-crown-6 and *N*-(2-bromoethyl)phthalimide giving compounds 2,2′-((1,4,10,13-tetraoxa-7,16-diazacyclooctadecane-7,16-diyl)bis(ethane-2,1-diyl))-bis(isoindoline-1,3-dione) (**1**) and 2-(2-(1,4,10,13-tetraoxa-7,16-diazacyclooctadecan-7-yl)ethyl)isoindoline-1,3-dione (**3**). Compound 2-(2-(16-ethyl-1,4,10,13-tetraoxa-7,16-diazacyclooctadecan-7-yl)ethyl)isoindoline-1,3-dione (**4**) was formed by process of alkylation of compound 2-(2-(1,4,10,13-tetraoxa-7,16-diazacyclooctadecan-7-yl)ethyl)isoindoline-1,3-dione (**3**) with ethyliodide. Similarly compounds 2,2′-((1,4,10,13-tetraoxa-7,16-diazacyclooctadecane-7,16-diyl)bis(ethane-2,1-diyl))bis-(isoindoline-1,3-dione) (**1**) and 2-(2-(16-ethyl-1,4,10,13-tetraoxa-7,16-diazacyclooct adecan-7-yl)ethyl)isoindoline-1,3-dione (**4**) were subjected to procedure of hydrazinolysis to yield compounds 2,2′-(1,4,10,13-tetraoxa-7,16-diazacyclo-octadecane-7,16-diyl)diethanamine (**2**) and 2-(16-ethyl-1,4,10,13-tetraoxa-7,16-diazacyclooctadecan-7-yl)ethanamine (**5**), respectively. Then reaction of a specific compound 3,4-diethoxycyclobut-3-ene-1,2-dione at different conditions like anhydrous EtOH at room temperature with $Zn(CF_3SO_3)_2$ and 4-trifluoromethylaniline resulted in compounds 3-((3,5-bis(trifluoromethyl)phenyl)amino)-4-ethoxycyclobut-3-ene-1,2-dione (**6**) and 3-ethoxy-4-((4-(trifluoromethyl)phenyl)amino)cyclobut-3-ene-1,2-dione (**7**) [53,57]. Finally, according to literature, reactions between compounds 2,2′-(1,4,10,13-tetraoxa-7,16-diazacyclooctadecane-7,16-diyl)diethane-amine (**2**), 2-(16-ethyl-1,4,10,13-tetraoxa-7,16-diazacyclooctadecan-7-yl)ethaneamine (**5**) and compounds 3-((3,5-bis(trifluoromethyl)phenyl)amino)-4-ethoxycyclobut-3-ene-1,2-dione (**6**), 3-ethoxy4 ((4(trifluoromethyl)phenyl)amino)cyclobut-3-ene-1,2-dione (**7**) in the presence of triethylamine (Et₃N) and ethanol (EtOH) at room temperature yielded in the ultimate aza-crown ether-squaramide conjugate compounds (**8**–**11**) (Scheme 5) [57,58].

### 3.4. Synthesis of Dibenzothiazolylodibenzo-18-crown-6 Ether

The reagents used to prepare diformyl dibenzo-18-crown-6 ether in the first step involved hexamine and trifluoroacetic acid. In the second step, the reagents used were 2-aminobenzenethiol in dimethylformamide without any catalyst with diformyl dibenzo-18-crown-6 ether by heating at 60 °C for 24 h which yielded a low amount of final product as 60%. As the yield came out significantly less, for this purpose, dimethylformamide was replaced with methanol which unexpectedly and finally resulted in a 90% yield of dibenzothiazolylodibenzo-18-crown-6 (Scheme 6) [53].

### 3.5. Synthesis of (γ-Arylpyridino)-dibenzoaza-14-crown-4 Ether

A fruitful procedure of domino-type condensation was carried out to synthesize (γ-arylpyridino)-dibenzoaza-14-crown-4 ether. The reaction temperature of 120 °C, reaction time of 6 h and three major components used for this type of condensation were: an aromatic aldehyde, ammonium acetate, and 1,5-bis-(2-acetylphenoxy)-3-oxapentane [59]. As γ-arylpyridine plays a significant role as a pharmacophoric fragment prompting an effective impact on the bioactivities of various drug molecules, which became a motivation for the synthesis of (γ-arylpyridino)-dibenzoaza-14-crown-4 ether (Scheme 7) [60,61].

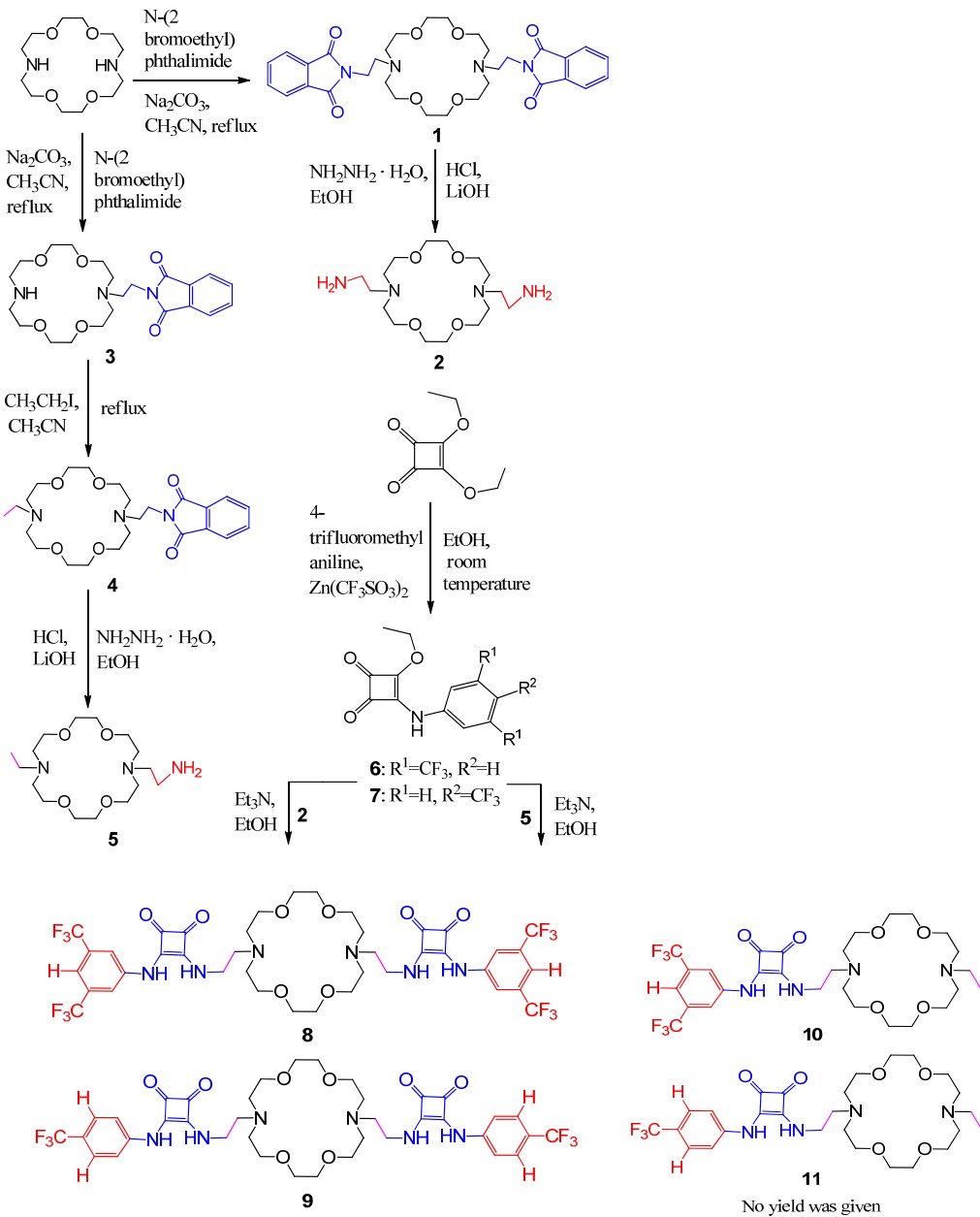

**Scheme 5.** Synthesis of aza-crown ether-squaramide conjugate compounds (**8–11**).

**Scheme 6.** Dibenzothiazolydibenzo-18-crown-6 ether reaction mechanism for synthesis.

**Scheme 7.** Formation of (γ-arylpyridino)-dibenzoaza-14-crown-4 ether.

## 4. Biological and Pharmacological Analysis of Crown Ethers

Crown ethers are found to have synergistic biological and pharmacological potential, they are widely used in drug delivery. The activities include anti-cancer, anti-microbial, anti-inflammatory and drug delivery.

### 4.1. Antibacterial Potential

The Schiff bases containing bis-benzo-15-crown-5 and their sodium derivatives were examined using a well diffusion method against bacterial strains; *Staphylococcus aureus*, *Shigella dysenteria* type 2, *Listeria monocytogenes*, *Escherichia coli*, *Salmonella typhi* H, *Staphylococcus epidermis*, *Brucella abortus*, *Micrococcus luteus*, *Bacillus cereus,* and *Pseudomonas putida* while choosing DMF as a control. The Schiff bases with 5-methoxy groups have shown maximum activity against strains with a concentration of $10^3$ μM [54]. Interestingly, the chelates (**12**, **13** and **14**) containing Na are found more potent against bacterial strains than bis-benzo-15-crown-5 compounds as shown in Figure 4. The bactericidal activity was compared with commercial drugs; kanamycin K30, sulfamethoxazole SXT25, Amoxycillin AMP10, sulbactam SCF, and nystatin NYS100 (Table 1) to describe the potency of compounds [54].

**Figure 4.** Structure of Schiff bases containing bis-benzo-15-crown-5 ether **12**–**14**.

**Table 1.** Comparison of antibacterial and antifungal activity of crown ethers and standard drugs (zone of inhibition in mm).

| Samples | *E. coli* | *S. typhi* H | *Br. abortus* | *L. monocytogenes* | *C. albicans* |
|---|---|---|---|---|---|
| **12** | 16 | 13 | 23 | 12 | 24 |
| **13** | 17 | 14 | 24 | 12 | 23 |
| **14** | - | 14 | 15 | 12 | 22 |
| **K30** | 25 | 20 | - | 15 | - |
| **AMP10** | 10 | 11 | - | 16 | - |
| **SXT25** | 18 | 17 | - | 11 | - |
| **SCF** | - | - | 12 | - | - |
| **NYS100** | - | - | - | - | 20 |

The novel chromene crown ethers and their respective sodium and potassium complexes were tested against strains; *S. dysenteria* type 2, *S. epidermidis*, *P. vulgaris*, *K. pneumonia* sp., *Shigella dysenteria*, and *S. marcescens* sp. Most of the derivatives were found to be moderately active against these pathogens. The compound **15** given in Figure 5. showed 15 mm zone of inhibition against *Shigella dysenteria* [62].

**15**

**Figure 5.** Structure of chromene crown ether complexed with the sodium ion **15**.

*4.2. Antifungal Potential*

The crown ethers, their complexes of sodium moieties along with their alkali metal complexes, and other crown ether derivatives were investigated against fungal strain *Candida albicans*, revealed moderate to significant antifungal activity when compared with positive controls [54,62]. The values are mentioned in Table 1.

Four novel 4,4′-di(2,2′-benzimidazolyl)dibenzo18-crown-6 (**16a–d**) were investigated for their antifungal efficacy against *Aspergillus* sp. using DMF and bavistin as control and standard. To study the antifungal activity, agar well diffusion method was chosen. The compound **16d** given in Figure 6. was found to be most active against fungal strains with maximum inhibitory activity (Table 2) [53].

**16d**

**Figure 6.** Structure of 4,4′-di(2,2′-benzimidazolyl)dibenzo18-crown-6 **16d**.

**Table 2.** Antifungal activity of 18-crown-6 and drugs.

| Studied Compounds | *Aspergillus niger* | *Aspergillus oryzae* |
|---|---|---|
| | **500 ppm** | **500 ppm** |
| **16d** | 24 | 28.5 |
| DMF | - | - |
| Bavistin | 33 | 22 |

*4.3. Anticancer Potential*

Cytotoxic potential of aza-crown ether-squaramide conjugates (**17a–d**) were measured through MTT assay at the concentration of 50 μM choosing three cancer cell lines; human lung carcinoma (A549), human breast cancer (MCF-7) and human liver cancer (HepG2). The inhibitory concentration (IC$_{50}$) in cell growth was compared with the doxorubicin drug. Results have shown that compounds **17b** and **17d** given in Figure 7. displayed moderate anti-proliferative activity as compared to others (Table 3). Furthermore, the compounds enhanced cytotoxicity in HeLa cells in the presence of chloride or sodium ions that move across the cell membrane and promote cell apoptosis [55]. Another study reported in vitro anti-proliferative potential of the crown ether acyl derivatives in HBL-100, HeLa, SW1573, and WiDr human solid carcinomas, using cisplatin and etoposide as positive controls. The derivative showed comparable activity against WiDr cell line [63]. Two aza-crown ethers, *N*,*N*'-bis (dithiocarbamate)-1,10-diaza-18-crown-6 (L$^{2-}$) were studied for their anticancer potential human cervix carcinoma cell line HeLa-229, the human ovarian carcinoma cell line A2780, and cisplatin-resistant mutant A2780 cis cells. The analysis revealed a new strategy to design metal-based drugs. The aza-crown ether Pt complex ligand increases affinity for antitumor activity (Table 3), whereas the addition of Na or K salts further boosts the activity as compared to cisplatin, taken as reference drug [64].

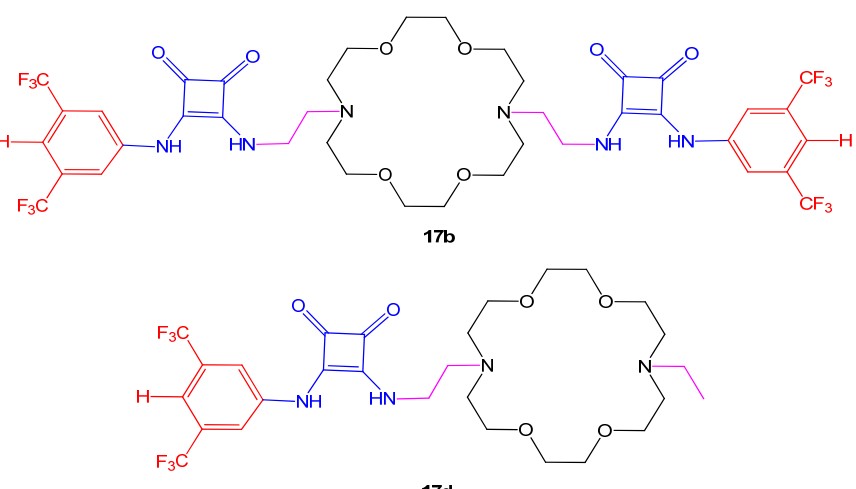

**Figure 7.** Structures of aza-crown ether-squaramide **17b**, **17d**.

**Table 3.** Anticancer potential of crown ethers and drugs in Hela cells.

| Studied Compounds | Hela Cell Line |
|---|---|
| **17b** | >50 |
| **17d** | >50 |
| Doxorubicin | 0.13 ± 0.01 |
| Aza-crown ether Pt complex | 6.4 ± 0.2 |
| Cisplatin | 0.53 ± 0.6 |

### 4.4. Toxicology

The earthworm *Eiseniafetida* was exposed to 18-crown-6 compounds with variable concentrations in the dry soil to investigate the toxic effect of components towards growth, respiration, and burrowing behavior of the worm in soil. The experiments revealed a reduced rate of worm growth, and burrowing behavior indicated by decreased mass index of the worm from (96.3 $\pm$ 5.6 to 78.2 $\pm$ 6.4) measured in mean $\pm$ SD, as the concentration of 18-crown-6 increased from 0 to 200 mg kg$^{-1}$ dry soil [65].

### 4.5. Amyloidogenesis Inhibitory Activity

The novel crown ether analogs were tested to evaluate their inhibitory potential against mutated transthyretin protein that promotes amyloidosis, forming amyloid fibril. Amyloid fibrils gather in specific tissues, including the eyes and the heart, and leave adverse effects. Anti-transthyretin amyloid genesis activities of crown ethers were observed using X-ray crystallography and binding assay using probes. Diflunisal, an anti-inflammatory drug was taken as a reference [66]. Among all, the 4′-carboxybenzo-18C6 (**18**) given in Figure 8. showed best anti amyloid fibril formation for transthyretin protein (Table 4).

**18**

**Figure 8.** Structure of 4′-carboxybenzo-18-crown-6 ether **18**.

**Table 4.** Amyloid genesis inhibitory potential of **18**.

| Studied Compounds | Concentrations | Anti-Amyloid Genesis Activity (%) |
|:---:|:---:|:---:|
| **Diflunisal** | 0.5 µM | 80 $\pm$ 9.0 |
| | 1.5 µM | 35 $\pm$ 1.0 |
| **18** | 0.2 mM | 99 $\pm$ 0.25 |
| | 0.6 mM | 92 $\pm$ 1.0 |

### 4.6. Electrochemical Sensor

Crown ether-based sensors were constructed to determine serotonin in human serum. Crown ether was mixed with carbon nanotubes-ionic liquid crystal and at glassy carbon electrode surface (GC/(CNTs-ILC) allowed a stable host-guest inclusion complex between crown ethers and neurotransmitters. The crown cavity of 18-Crown-6 allows H-bond formation with the indole ring of serotonin. The sensor enables to examine drugs at a low cost [67]. Furthermore, potassium ion (K$^+$) microsensors were made using electrochemical impedance spectroscopy (EIS) to understand potassium ions transport across the cellular membrane. The building block of sensors was 1-aza-18-crown-6 functionalized graphene oxide (Crown-GO). The crown ether part captures potassium ions, whereas graphene oxide provides the binding ability [68]. A study reports K$^+$ chemical sensor development based on a self-assembled monolayer of 4-aminobenzo-18-crown-6 ether as selective ionophore that shows reproducible activity in disease diagnosis [69]. The 4-aminobenzo-18-crown-6 modified gold nanoparticles have been used by researchers to make the sensor to detect K$^+$ in human urine samples through colorimetric assay. The K$^+$ detection was compared with few other ions through UV-Vis and showed excellent anti-interference activity [70].

### 4.7. Disposable Potentiometric Sensor

Dibenzo 24-crown-8-ether as shown in Figure 9. based potentiometric sensors were built to determine Biperiden hydrochloride in urine and plasma. Biperiden is used to treat Parkinsonism. The findings demonstrated improved response time, lifetime, sensitivity, selectivity, and the possibility of miniaturization of the sensor compared with γ-cyclodextrins, calixarenes, and buckminsterfullerene $C_{60}$ [71].

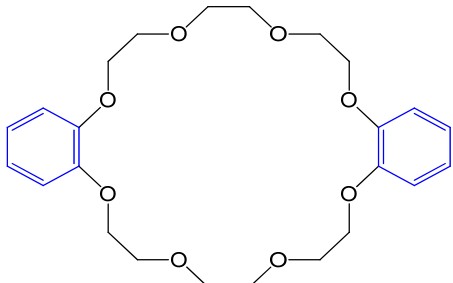

**Figure 9.** Structure of dibenzo-24-crown-8-ether.

### 4.8. Chiral Catalyst

Crown ethers are reported to have prime importance in performing the catalytic activity in many phase transfer reactions. The monosaccharide-based crown ethers shown in Figure 10. have been used in chacones epoxidation, leading to understanding the enantioselectivity of compounds that play a key role in drug designing [72].

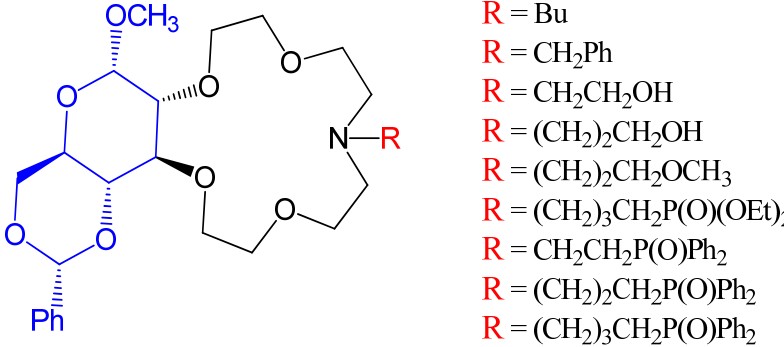

R = Bu
R = $CH_2Ph$
R = $CH_2CH_2OH$
R = $(CH_2)_2CH_2OH$
R = $(CH_2)_2CH_2OCH_3$
R = $(CH_2)_3CH_2P(O)(OEt)_2$
R = $CH_2CH_2P(O)Ph_2$
R = $(CH_2)_2CH_2P(O)Ph_2$
R = $(CH_2)_3CH_2P(O)Ph_2$

**Figure 10.** Monoaza-15-crown-5 lariat ether containing *D*-glucose.

### 4.9. Peptide Interaction and Self-Assembly

A neutral antimicrobial peptide holding crown ether side chain has been modified by interacting with cationic arginine to form a secondary structure, which interacts with the lipid membrane of the cell. The study finds its potential in the pharmaceutical industry [73]. Due to holding particular multi-cavity structures and diverse complexity, the iptycene-derived crown ether serves as first generation of synthetic hosts in molecular recognition. It can be a promising candidate for constructing self-assemblies. Hence, provide broader spectra of applications in biological and material sciences.

### 4.10. Crown Ethers as P-Glycoprotein Inhibitors

The effect of monoaza- and diaza-18-crown-6 ethers was studied as multidrug-resistant (MDR) reversal in model cell lines. The compounds were found to be more active P-glycoprotein (P-gp) inhibitors and increased apoptotic activity than verapamil, a commercially available drug. The activity was performed through ATPase activity showed that crown ethers inhibit P-gp without affecting their locality [74].

### 4.11. DNA Targeting

A G-quadruplex DNA (G4-DNA) probe with far-infrared luminescence property was synthesized and characterized by NMR and MS using modified aza-crown ether with triphenylaminequinoline derivative (TPAQD-ACE). The metal conjugate probes of TPAQD-ACE showed different binding abilities towards DNA, among which Ni and Fe presented maximum signaling at 640 nm in buffer. Furthermore, the cell staining results assured that Nickle metal ion probes bind more efficiently to DNA present in cells. The application has drawn significant attention of researchers in the biological field [75].

### 4.12. Enzyme Activation

The effect of crown ether ligands (**19–27**) shown in Figure 11. was studied on enzyme activities of hCA (human carbonic anhydrase) purified from erythrocytes using SDS-PAGE. All analogs activated the enzyme activity except **22** and **23** that interacted with the active part of the enzyme and inhibited the interaction between enzyme and substrate. The inhibitory potential of **22** and **23** was compared to Acetazolamide. Enzyme activity was increased with an increase of the ring cavity [76].

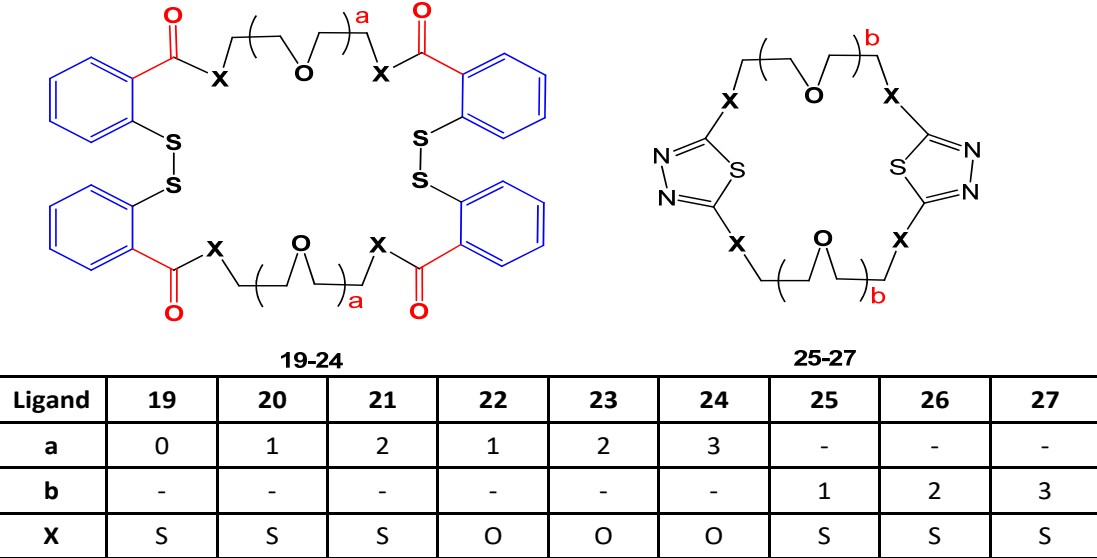

| Ligand | 19 | 20 | 21 | 22 | 23 | 24 | 25 | 26 | 27 |
|--------|----|----|----|----|----|----|----|----|----|
| a | 0 | 1 | 2 | 1 | 2 | 3 | - | - | - |
| b | - | - | - | - | - | - | 1 | 2 | 3 |
| X | S | S | S | O | O | O | S | S | S |

**Figure 11.** Structures of crown ethers (**19–27**).

### 4.13. Crown Ethers as Vesicles

In another study, Benzo-15-crown-5active polydiacetylene (PDA) based vesicle receptors were investigated to detect metal ions. The receptors showed a strong interaction with lead ions ($Pb^{2+}$) visualized as color change from blue to red in buffer (pH = 7.2) and further characterized through UV spectroscopy. The maximum interaction of $Pb^{2+}$-crown ethers on lipid interface makes crown ethers unique to detect biomolecules like proteins, sugars, and microbes [77].

### 4.14. Membrane Anchors

Lipid-nucleic acids (LiNAs) conjugates based on polyaza crown ethers were synthesized to investigate the fusion of liposomes to enhance content mixing within the cell. LiNAsget anchored into the outer liposomes leaflet and provide strong fusogens that may work as a key tool for fusing cell membranes [75].

### 4.15. Transfection Activity

The complex formation with DNA of two nitrogen-pivoted aza-crown ethers linked to the cholesteryl-fused ring system *N*-(cholesteryloxycarbonyl)aza-15-crown-5 and *N*-(cholesteryloxycarbonyl)aza-18-crown-6incorporated to liposomes were observed in the

human embryonic kidney cell line (HEK293). The increased transfection activity revealed stable DNA-protective lipoplexes containing aza-crown ethers, attributed to perturbations of the endosomes and the loosely packed cargo plasmid DNA [78].

### 4.16. Crown Ethers-Tyrosine Kinase Inhibitors

Crown ethers (CEs) fused with quinazoline were studied as epidermal growth factor receptor (EGFR) inhibitors via in vitro tyrosine-kinase and phosphorylation assays. Some of these compounds are potent inhibitors of EGFR and give a broad spectrum of human tumors inhibition [79].

### 4.17. Fluorescent Chemosensor

Crown ether-acylhydrazone (**L**) based unique chemosensors were built in methanol solution to determine fluorescence of different metal ions. An unusual observation for $Al^{3+}$ ions was seen in UV, a color change from pale blue to bright blue at 444 nm reveals an intense correlation of crown ethers and $Al^{3+}$. Figure 12. shows the binding interaction between **L** and $Al^{3+}$. These chemosensors are in progress to find applications in biochemistry [68].

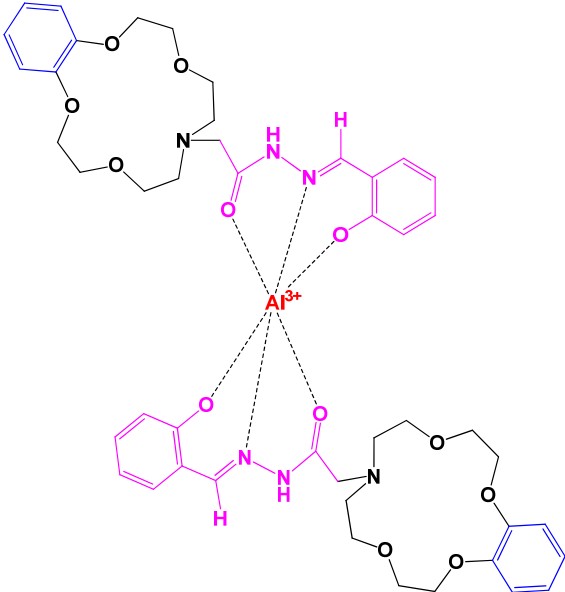

**Figure 12.** Crown complex showing the interaction between **L** and $Al^{3+}$.

### 4.18. Ion Transport

Synthetic bucky ball-based molecular balls were designed linked with CEs that enable ion transport across the biological membrane. Generally, the molecular balls linked to small CEs showed high selectivity towards ion transport compared to those containing larger CEs [80].

### 4.19. Computational Study

Lithium-ion binding to 8-Crown-4 was investigated computationally to verify the ion binding using Spartan 10 software, and further optimization was examined using Gaussian 09. DFT computations showed two stable conformations of 8-Crown-4, the crown (Cr) and boat-chair (BC). The free energy level indicates that the BC conformer is intrinsically more stable than the Cr conformer. However, in nitromethane, the Cr conformer appeared more stable than the BC due to a 2.4 kcal mol$^{-1}$difference in solvation energy [81]. A molecular docking study investigated the binding mode between TPAQD-ACE and the human G-quadruplex using AutoDock Vina. The results inferred that the introduction of metal ions to the compound enhances the fluorescent signal intensity of TPAQD-ACE and promotes the binding abilities of TPAQD-M-ACEs to G4 DNAs [82]. The Na$^+$ and K$^+$

selectivity of 18-Crown-6 ether, dibenzo-18-crown-6, and cryptans was compared through computational study. The results showed more affinity of compounds towards $K^+$ than $Na^+$ due to the solvent effect [83].

### 4.20. Antimicrobial Peptides

Antimicrobial peptides (AMPs) are constituted of numerous species, immune systems and minimize the development of bacterial resistance. Different factors such as length, hydrophobicity, amphiphilicity, flexibility, and the net charge on AMPS control its selectivity and potency. This 14-residue peptide as shown below in Figure 13. exhibits moderate permeability across EYPC vesicles and contains four 21-crown-7 modified phenylalanines with 10 leucines. The peptide 12-residue exhibits high permeability where an increase in length up to 13–16 residues enhances activity showing smaller ring size may decrease peptide activity [84].

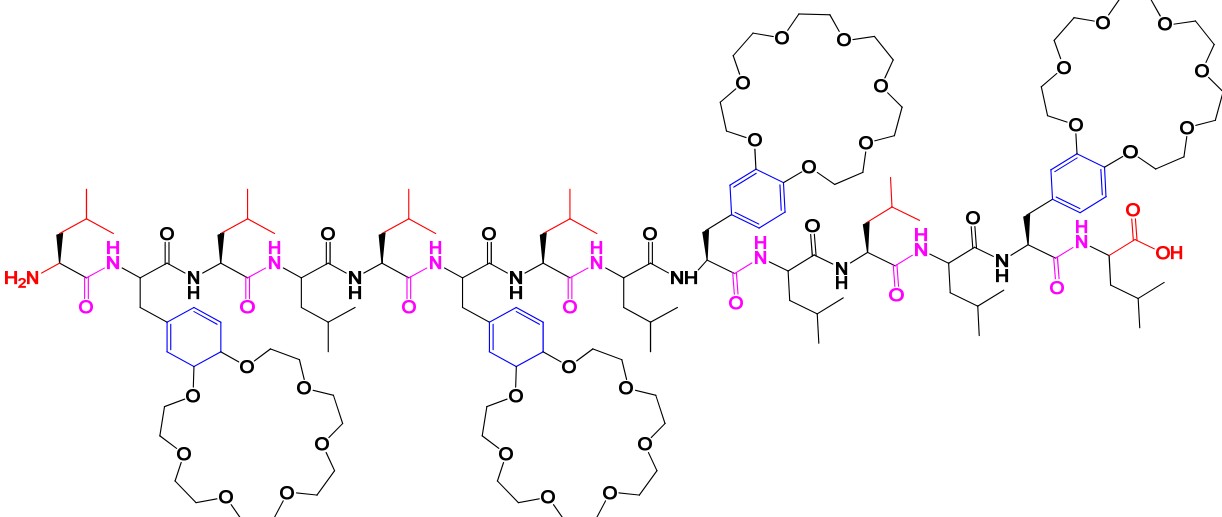

**Figure 13.** Structure of 14-residue model peptide.

### 4.21. Covalent Organic Frameworks (COFs)

COFs periodic structure potential for catalysis, energy storage, ions conductor, gas separation, and sorption. Py-$B_{24}C_8$-COF andPy-$B_{18}C_6$-COF exhibit capture capacity capture $Cs^+$ and $K^+$, respectively. COFs offer's platform for precisely designing ions absorbents [85]. Olefin Chiral COFs(CCOFs 17-R and 18-R) exhibit higher enantioselectivity than their reduced Chiral COFs structures when used as fluorescence sensors to detect chiral amino alcohols [86].

### 4.22. Quinolines Derivatives

Many quinolines derivatives possess biological and physiological properties. Their antioxidant, antimicrobialand cytotoxic effects regarding breast tumor cells were reported. Crown ethers derived quinoline as shown in Figure 14. were biologically inactive against Gram-positive and Gram-negative bacteria. Moreover, they exhibited a low cytotoxic effect against breast tumor cells [87].

**Figure 14.** Structure of crown ethers derived quinoline.

### 4.23. Piperidones

Piperidones exhibit biological activities as anticancer, anti-inflammatory, antimicrobial, and anti-Alzheimer agents.Introduction of piperidone into crown ethers thataffectbioactivity while its derivatives exhibit antimicrobial and ion selectivity properties. Newly synthesized $2^3$-methyl-4,7,10-trioxa-2(2,6)-piperidina-1,3(1,2)-dinaphthalenacyclodecaphan-24-one azacrown ether was isolated in 45% yield as a final product shown in Scheme 8. shows in-vitro cytotoxic effects against human cancer cell lines as rhabdosarcoma (RD), human lung adenocarcinoma (Lu1), human breast adenocarcinoma (MCF-7), and hepatocellular carcinoma (Hep-G2). This azacrown ether exhibited antimicrobial activity against *Pseudomonas aeruginosa*, *Staphylococcus aureus* subsp. Aureus, *Fusarium oxysporum*, *Saccharomyces cerevisiae*, *Candida albicans*, *Aspergillus niger* and *Bacillus subtilis* [88].

**Scheme 8.** Synthesis of $2^3$-methyl-4,7,10-trioxa-2(2,6)-piperidina-1,3(1,2)-dinaphthalenacyclodecaphan-24-one aza crown ether.

### 4.24. Dibenzothiazolyldibenzo-18-crown-6 Ether

Structure of dibenzothiazolyldibenzo-18-crown-6 ether given in Figure 15. shows antimicrobial activity. It exhibited activity against (Gram +ve) bacteria *S. aureus* and antifungal activity against *A. niger*. Studies revealed that it is ineffective against *C. albicans* fungus and bacteria *E. coli*. In addition to this, it can effectively determine trace levels of palladium but is ineffective in other metal ions presence [62].

**Figure 15.** Structure of dibenzothiazolyldibenzo-18-crown-6 ether.

*4.25. Benzo-15-crown-5 Substituted Coumarin*

Benzo-15-crown-5 substituted coumarin and their derivative compounds can be used as chemosensors selective fluorescent $Cu^{2+}$ and $Fe^{3+}$. They exhibit antifungal and antibacterial activities against both gram-positive and gram-negative bacteria. They are found more effective than commercial antifungal agents and antibiotics.

*4.26. (ɤ-Arylpyridino)-dibenzoaza-14-crown-4 Ethers*

They possess activity against tumor cell lines. Compounds **28(a–h)** as shown below in Figure 16. possess antineurotoxic, antineoplastic and cardioprotective activities. In vitro cytotoxicity tests reveal that **28c** inhibits human rhabdomyosarcoma (RD), **28d** inhibits human rhabdomyosarcoma (RD), human breast adenocarcinoma (MCF7), human uterine (FL), human hepatocellular carcinoma (HepG2). **28a** does not show scavenging activity for free radical [53].

**28(a-h)**

**a:** R = H, **b:** R = C4-OMe, **c:** R = C2-OH, **d:** R = C4-Cl
**e:** R = C4-NO$_2$, **f:** R = C4-OH, **g:** R = C4-Br, **h:** R = C2-Cl

**Figure 16.** Structure of (ɤ-arylpyridino)-dibenzoaza-14-crown-4 ether **28(a–h)**.

*4.27. Benzo-Oxo Crown Ethers and Macrocyclic Benzothio Crown Ethers*

These CEs given below in Figure 17. showed high selectivity for $Fe^{3+}$ as compared to $Ag^+$, $Pb^{2+}$, $Co^{2+}$, $Cd^{2+}$, $Zn^{2+}$, $Ca^{2+}$, while no selectivity was observed for $Na^+$, $K^+$ and $Li^+$ ions. These ligands can be used as enzyme inhibitors, metal sensors, and antimicrobial/antifungal agents [89].

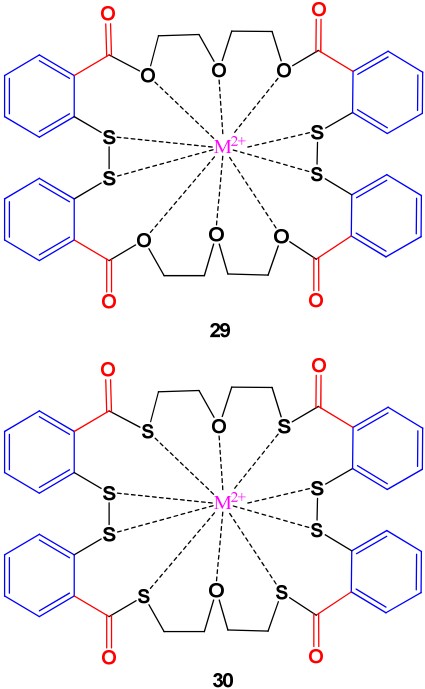

**Figure 17.** Structure of benzo-oxo crown ether (**29**) and macrocyclic benzo-thio crown ether (**30**).

*4.28. 4-((Substituted bis-indolyl)methyl)-benzo-15-crown-5 Ether*

Derivatives of these crown ethers were filtered, prepared with 56% yield at 260 °C melting point and their purity was checked by TLC. Out of these, the one given below in Figure 18. has the ability to detect $Hg^{2+}$ ions in an aqueous medium [90].

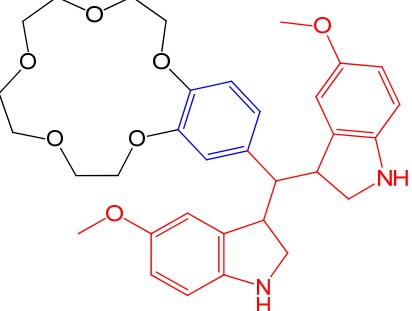

**Figure 18.** Structure of 4-((Substituted bis-indolyl)methyl)-benzo-15-crown-5 ether.

## 5. Conclusions

Crown ethers are extremely interesting and important in the fields of chemistry, materials science, separation, biology, catalysis, transport and encapsulated processes. CEs are extensively used in drug delivery due to enzyme regulation, interactions with DNA, and anti-microbial activity. CEs have shown potential to interact with enzymes, such as R-chymotrypsin, lipases, and subtilisin Carlsberg. Ionophoric properties allow CEs to transport through membranes and interact with the living system. CEs generate complexes by firmly binding certain cations. The oxygen atoms are ideally positioned to coordinate with a cation located in the ring's core. Crown ethers are valuable in phase transfer catalysis because they form salts that are soluble in nonpolar solvents. Stability constants can be improved by the selection of CEs for various ions and via the extraction method. Moreover, the crowns have been entirely used to produce ion sensors that identify the presence of target ions. Another significant feature of these compounds is forming selective and stable complexes with various organic and inorganic cations. In a nutshell, CEs use as anticancer, anti-inflammatory, antibacterial, and antifungal agents and have explored their amyloid

genesis inhibitory activity. Crown ethers are a promising and rising class of chemicals that may have a strong foothold in the biological sciences as well. We are certain that future generations of crown ethers will create new and inventive applications, maybe as novel anticancer treatments for many diseases.

**Author Contributions:** Conceptualization: F.U., M.M. (Moriusz Mojzych); data collection: F.U., T.A.K., J.I., S.A., M.R.K., S.U.; Analysis of results: M.F.A.K., M.M. (Moriusz Mojzych), M.F.u.R.; writing—original draft preparation: F.U., T.A.K., S.A.; writing—review and editing: K.K.-M., M.M. (Muhammad Mustaqeem) All authors have read and agreed to the published version of the manuscript.

**Funding:** This research received no external funding.

**Institutional Review Board Statement:** Not applicable.

**Informed Consent Statement:** Not applicable.

**Data Availability Statement:** Data sharing not applicable.

**Acknowledgments:** Not applicable.

**Conflicts of Interest:** The authors declare no conflict of interest.

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
