# Peer review of "Heterocyclic Crown Ethers with Potential Biological and Pharmacological Properties: From Synthesis to Applications"

_applsci, doi:10.3390/app12031102_

Round 1

Reviewer 1 Report

This is a review of the chemistry of crown ethers.  I am not adequately familiar with the literature (and there is a lot) on this subject to judge how complete it is, but there is a narrative and story that is told.

Just a trivial thing

Page 5

"The compound 1,2-bis(bromomethyl)benzene in EtOH was reacted with hydroxyl benzaldehydes in the presence of NaOH"   This is poorly worded.  They reacted on their own when they were mixed together.    

Author Response

Page 5
1)      Synthesis: “The compound 1,2-bis(bromomethyl)benzene in EtOH was reacted with hydroxyl benzaldehydes in the presence of NaOH.” This is poorly worded. They reacted on their own when they were mixed together.

Response: We thank reviewer for this precious suggestion. Wording corrected as suggested.

Reviewer 2 Report

Manuscript Number: applsci-1496161
entitled: Heterocyclic Crown Ethers with Potential Biological and Pharmacological properties: from Synthesis to Applications

This is an interesting scientific study. Therefore, the manuscript is suitable for publication in Applied Science after considering the below comments:

  1. Please change “Scheme 1.” Please draw a general structure (-s) of Crown ethers (CEs) and a general symbol of metal cations discuss in this manuscript. The present arrows mean mesomeric interaction between the presented structures, so please modify it.
  2. It is correct that “These are extremely versatile compounds exhibiting higher binding affinity towards metal ions, including s-block and transition metal ions.” So it will be nice to present equation 2M(s) + L → M+L + M- from (page 2; ACCOUNTS OF CHEMICAL RESEARCH / VOL. 31, No. 2, 1998, pp55-61).
  3. I recommended the rev. by Gokel, et al. Crown Ethers: Sensors for Ions and Molecular Scaffolds for Materials and Biological Models Rev. 2004, 104, 2723-2750. It should be helpful during revising this manuscript.
  4. Besides nonactin, there is another important biologically active compound, Valinomycin. This ionophore is highly flexible and possesses multiple donors.
  5. To every Schemes presented, a synthesis, please add the yields of products, all conditions, e.g., temperature. Additionally, the isolation procedures.
  6. Would you please add space to 60space°C and passim.
  7. “Biological & Pharmacological analysis of crown ethers” in this chapter, the mechanism of Crown Ethers action will be very welcome.
  8. Figure 4. Would you please add the charge of sodium cations?
  9. The title of Figure 13. “Structure of peptide” should be changed. The presented structure is not the general structure of peptides but a particular one.
  10. Page 18, “Derivatives of these crown ethers were prepared in good yield,” please give exact data.
  11. Conclusion part. The conclusion section is too short and unsatisfactory and does not indicate which could have been the real motivation for inspiring this work. The few sentences forming this section do not add anything to the significance of this compendium and do not offer any justification for this review.
  12. In the References part, please add all authors names, not et al. Ref.1. and 24,
  13. Would you please add the titles of any ref?
  14. Fragmentary Ref.5, 6, 7, 8, 10, 13, 22, 43, 50, and so on.
  15. Ref. 38, 42 change capital letters to small letters.

Author Response

We thank the reviewer for your precious suggestions. 

Ad 1 - Ad 3
As suggested the required modification done.

4) History: Besides nonactin there is an important biologically active compound, Valinomycin. This ionophore is highly flexible and possesses multiple donors.

Response: As suggested the required modification done.

5) Synthesis: To every schemes presented, a synthesis, please add the yields of products, all conditions, e.g., temperature, Additionally, the isolation procedures.

Response: As suggested the required modification done.

6) Synthesis: Add space to 60space°C and passim.

Response: As suggested the required modification done.

7) Biological activities: Figure 4. Add the charge of sodium cations.

Response: As suggested the required modification done.

8) Biological activities: The title of Figure 13. “Structure of peptide” should be changed. The presented structure is not the general structure of peptides but a particular one.

Response: As suggested the required modification done.

9) Biological activities: Page 18, “Derivatives of these crown ethers were prepared in good yield,” give exact data.

Response: As suggested the required modification done.

10) Conclusion:

Response: As suggested the required modification done.

11) Reference:

Response: As suggested the required modification done.

12) Reference:

Response: As suggested the required modification done.

13) Reference:

Response: As suggested the required modification done.

14) Reference:

Response: As suggested the required modification done.

Reviewer 3 Report

This manuscript is a review on heterocyclics crown ethers describing there synthesis and applications. It covers the chemical structure, synthesis and application of same.

In general, manuscript content is ok for this journal. However, language must be thoroughly improved. Content can be elaborated for each section.

Table providing the summary of the literature on heterocyclics, use, and synthesis method will be helpful.

I believe the content will interesting to readers, but major changes are need in revising the language, and elaboration of each section. some sections have only few lines, and single references. If not may references are available then it can me merged with other section, instead of separate classification.   

Author Response

We express our gratitude to the reviewer for all valuable comments.

  • Language must be thoroughly improved and content can be elaborated.

Response: As suggested the required modification done.

  • Table providing the summary of literature on heterocycles, use and synthesis method will be helpful.

Response: As suggested the required modification done.

  • Some sections have few lines and single references. They should be merged with other sections instead of separate classification.

Response: As suggested the required modification done.

Reviewer 4 Report

The paper consists of a comprehensive review on Crown ethers (CEs) including general description, historical data, synthetic information and specific applications. The manuscript is well written and could be useful for the general readers offering a wide view of the reviewed topic. A further improvement could be achieved separating from the 4. Biological & Pharmacological analysis of crown ethers section, as a different section, the general sensor/analytical/catalytic applications (4.6. Electrochemical sensor, 4.7. Disposable potentiometric sensor, 4.8. Chiral catalyst, 4.17. Fluorescent Chemosensor, 4.21. Covalent Organic Frameworks (COFs)).

Minor issues

Line 88

Pederson found that this isolated by-product from a potent complex with K+  ions.

Should be

Pederson found that this isolated by-product form a potent complex with K+ ions.

In Fig. 11 A and B in the table should match a and b in the structures (both capital or both lowercase)

Author Response

Dear reviewer, we have made required modification as suggested. 

Thanks for you comments.
Regards

Round 2

Reviewer 2 Report

Manuscript Number: applsci-1496161

entitled: Heterocyclic Crown Ethers with Potential Biological and Pharmacological properties: from Synthesis to Applications

The ms got somewhat better after the first round of revision.

However, I have the feeling that too many suggestions remained unanswered.

Would you please modify Scheme 1. Now there is „Binding of K+ by 18-crown-6 ether; K is complex stability constant [8]”; however, there is M+ instead of K+ on the graphic. I suggest modifying the text to „Binding of M+ by 18-crown-6 ether; K is complex stability constant [8]”.

Page 3, line 113, two dots. Typos

Scheme 3, 4, 5, 6, 7, 8, 18, and passim. Still, there are no yields of products on graphics (Schemes). This option should help readers to find such information easily. If the Authors do not mention the yields of obtained compounds, please write this, e.g., „no yields was given.”

On the final structure on Scheme 4, there are bonds between C-C instead of C-OCH3. Would you please modify it?

Page 15, Fig. 10. Would you please modify R meaning? For example, for me, the phosphorus atom is connected with 4 (CH2) units, two phenyls, and oxygen (P=O), in (CH2)4P(O)Ph2. Maybe just (CH2)3CH2P(O)Ph2, and so on?

Page 19, Figure 14. Possible should be quinoline instead of quinolone? And on the upper structure right quinoline fragment should possess a double bond instead of a single? Would you please explain what means „X”?

Fig. 14. Possible should be „Structures of crown ethers derived quinoline.”

Scheme 8. Synthesis of aza crown ether. Would you please name the presented structure?

Figure 16. Would you please modify the graphic? Now there are R = 2-OH, or 4-Cl, and so on. It should be R = C2-OH or C4-Cl, and so on.

Every scheme or Figure should be in the text, for example, Fig. 18.

Ref. 51. „Methods, T.; Sorption, D.V.; Diffraction, X.P.; Methods, C.; Sizing, P.; Amplitude, E.S.; Potential, Z. Physical 699 Chemistry Methods Physical Chemistry Methods. J. Phys. Chem. 1976, 80.” Would you please explain „Methods, T.; Sorption, D.V.; Diffraction, X.P.; Methods, C.; Sizing, P.; Amplitude, E.S.;” are these Authors? I was unable to find this reference. There are no pages.

Ref. 5 Why the Authors use bold letters for„J. Appl. Chem. 2017, 3, 237–244..”. It should be one dot.

Author Response

Round 2.

Reviewer 2
1) Would you please modify Scheme 1. Now there is „Binding of K+ by 18-crown-6 ether; K is complex stability constant [8]”; however, there is M+ instead of K+ on the graphic. I suggest modifying the text to „Binding of M+ by 18-crown-6 ether; K is complex stability constant [8]”.

Response: Above mentioned mistake was corrected.

2) Page 3, line 113, two dots. Correct typos.

Response: Corrected as recommended.

3)  In all synthesis schemes, yields of final products should be mentioned.

Response: Above mentioned mistake was corrected in the manuscript as suggested.

4) On final structure in Scheme 4, there are bonds between C-C instead of C-OCH3. Modify it.

Response: Corrected as recommended.

5) Page 15, Fig. 10. Modify the meaning of R.

Response: Above mentioned mistake is addressed in the manuscript as suggested.

6)  Fig. 14. Possible should be quinoline instead of quinolone. And on upper structure right quinoline fragment should possess a double bond instead of single. Also explain what is X.

Response: Corrected as recommended.

7) Fig. 14. Possible should be “ Structures of crown ethers derived quinoline.”.

Response: Above mentioned mistake is corrected in the manuscript as suggested.

8) Scheme 8. Synthesis of aza crown ether. Give the name of presented structure.

Response: Corrected as recommended.

9) Fig. 16. Modify the graphic. Instead of R=2-OH, or R=4-Cl and so on it should be R=C2-OH or R=C4-Cl and so on.

Response: Above mentioned mistake is corrected in the manuscript as suggested.

10) Every scheme or figure should be in text, for example, Fig. 18.

Response: Corrected as recommended.

11) Ref. 51. „Methods, T.; Sorption, D.V.; Diffraction, X.P.; Methods, C.; Sizing, P.; Amplitude, E.S.; Potential, Z. Physical 699 Chemistry Methods Physical Chemistry Methods. J. Phys. Chem. 1976, 80.” Would you please explain „Methods, T.; Sorption, D.V.; Diffraction, X.P.; Methods, C.; Sizing, P.; Amplitude, E.S.;” are these Authors? I was unable to find this reference. There are no pages.

Response: Above mentioned mistake is corrected in the manuscript as suggested.
12) Ref. 5 Why the Authors use bold letters for„J. Appl. Chem. 2017, 3, 237–244..”. It should be one dot.

Response: Corrected as recommended.

Reviewer 3 Report

accept

Author Response

This manuscript is a review on heterocyclic crown ethers describing their synthesis and applications. It covers the chemical structure, synthesis and application of same.

In general, manuscript content is ok for this journal. However, language must be thoroughly improved. Content can be elaborated for each section.

Response: As suggested the required modification done.

Table providing the summary of literature on heterocyclics, use and synthesis method will be helpful.

Response: As suggested the required modification done.

Some sections have few lines and single references. They should be merged with other sections instead of separate classification.

Response: As suggested the required modification done.

Round 3

Reviewer 2 Report

Dear Authors,

The authors conducted important data. This is an interesting paper. The present version of the manuscript is very well developed. The results contextualization and the manuscript are well written. The current version is much better; therefore, the manuscript is suitable for publication in the present form. Please only include one correction.

Scheme 7. Still, the substituent „R is not explained. Would you please add this information to the title, e.g., „Formation of (γ-arylpyridino)-dibenzoaza-14-crown-4 ether. R = ??” On the graphic, please add a space. It should be 120 space°C.